# Polyadic synapses introduce unique wiring architectures in T5 cells of *Drosophila*

**Eleni Samara**[ID]¤*, **Alexander Borst**

Department of Circuits-Computation-Models, Max Planck Institute for Biological Intelligence, Am Klopferspitz 18, Planegg, Germany

¤ Current address: Institute for Translational Neuroscience, Department of Neuroscience, NYU Grossman School of Medicine, 435 East 30th Street, 10016 New York NY, USA

* eleni.samara@nyulangone.org

## Abstract

Connectomes provide neuronal wiring diagrams and allow for investigating the detailed synaptic morphology of each connection. In the visual system of *Drosophila*, T5 cells are the primary motion-sensing neurons in the OFF-pathway and receive dendritic input from the excitatory Tm1, Tm2, Tm4, Tm9 and the inhibitory CT1 neurons. This connectivity, however, has not yet been investigated with respect to polyadic synapses known to be abundant in the fly nervous system. In this study, we use the FlyWire database and identify that Tm and CT1 cells wire on T5a dendrites via eight polyadic synapse types. We then explore the distribution of the different synapse types on T5a dendrites and find differences in their spatial patterns. Finally, we show that the polyadic morphology is setting a directional wiring architecture at the T5 network level. Our work showcases the directionality that polyadic synapses introduce in T5 connectivity.

## Introduction

One of the major goals in Neuroscience is to dissect structure-function relationships in neuronal circuits. This goal can be reached more easily by gaining access to brain-wide neuronal connectivity, namely the connectome. Recently, a female adult brain wiring diagram of *Drosophila melanogaster* was completed from electron-microscope reconstructions [1–4]. The dataset offers morphological access to approximately $10^5$ neurons and provides information on their connectivity via approximately $10^8$ chemical synapses. Interestingly, the majority of these chemical synapses are polyadic, i.e., one presynaptic site forms connections with multiple postsynaptic sites. Therefore, polyadic synapses introduce a diverging connection at the single bouton level.

Polyads are widely encountered in invertebrates [5], including the nematode *Caenorhabditis elegans* [6], the chordate *Ciona intestinalis* [7] and the annelid *Malacoceros fuliginosus* [8], and are also found in vertebrates as in the cat retina [9]. In *Drosophila melanogaster*, the polyadic chemical synapse morphology prevails in the

**Data availability statement:** The data are available at Zenodo: https://doi.org/10.5281/zenodo.17221018.

**Funding:** This work was funded by the Max Planck Society. The funders had no role in study design, data collection and analysis, decision to publish, or preparation of the manuscript.

**Competing interests:** The authors have declared that no competing interests exist.

early visual processing center of the brain, the optic lobe. While the role of polyads in synchronizing the postsynaptic neurons was suggested in *C. elegans* [6], the functional implications of polyadic morphology in visual circuits of the fruit fly still remain unknown.

Motion detection is a fundamental visual modality for the navigation of *Drosophila*, corroborated by the detailed study of circuits that compute directional information from either luminance increments (ON-motion pathway) or luminance decrements (OFF-motion pathway) for the past two decades [10–13]. T5 bushy cells are the elementary motion detectors in the OFF pathway and exist in four morphologically distinct subtypes a, b, c and d, where each subtype has its dendrite oriented into one of the four cardinal directions [14]. Aligned with their morphology, each subtype responds to either front-to-back, back-to-front, upward or downward motion, respectively [12]. T5 directional responses arise from the spatial and temporal offset observed among their cholinergic Tm1, Tm2, Tm4 and Tm9 and GABAergic CT1 input neurons [15–24] and the nonlinear interactions between them [25–27]. Nonetheless, the role of polyadic synapses in T5 wiring and consequently in T5 computation has not been addressed so far.

The categorization of dendritic spines to thin, branched, mushroom and stubby in vertebrates lead to the identification of their unique functional roles [28–31]. Similarly, categorizing morphological traits of polyadic synapses might reveal novel functional contributions. As a first step towards that goal, we study the polyadic synapse morphology in Tm1-, Tm2-, Tm4-, Tm9- and CT1-to-T5a connections and identify their wiring via different polyadic types. We then explore the T5a dendritic distributions of the different polyadic types. Finally, we show that polyadic synapses introduce a directional wiring motif at the T5 network level.

## Materials and methods

### Synapse proofreading

This paper analyzes an online open-access dataset via the FlyWire interface (flywire.ai) [1–4]. Information regarding the used neuronal IDs is reported in S1 Table. A subsequent version to the 783 public release dataset (FAFB) version was analyzed via FlyWire. The upstream circuitry (Tm1, Tm2, Tm4, Tm9, CT1) of five reconstructed T5a dendrites from the dorsal part of the right optic lobe were identified via Codex (synapse threshold≥5) as well as by the 'Fly connectivity viewer' in the FlyWire interface (no synapse threshold). Our analysis included neuronal inputs identified by both approaches. Codex cell type annotations were used together with morphological information [14–16,22] for the final cell type identification. Neuronal reconstructions were primarily performed by FlyWire users.

Synapse number Buhmann predictions [32] between the T5a neuron of interest and its respective input neurons were acquired from FlyWire's 'Fly connectivity viewer'. Cleft score was set to 50, so as to eliminate synaptic redundancy resulting from the combination of synaptic connections between the same neurons when their presynaptic locations are within 100nm$^2$. Then, we manually proofread chemical synapses based on the existence of four morphological markers: a. synaptic vesicles, b.

protein dense T-bar structure, c. synaptic cleft, and d. postsynaptic densities (or postsynaptic domains). Synapse coordinates (x,y,z) were allocated to the presynaptic T-bar and then compared to the automatic synapse coordinates from Codex corresponding to postsynaptic sites. In cases of multiple release sites (active zones) each identified with a T-bar [33], only the coordinates of one T-bar were included in our dataset (as in S1C Fig). Complex synaptic cases and their proofreading outcome are extensively described in S1A-D Fig. Most orphan twigs (non-traced dendritic spines) belonging to the same postsynapse with the T5a dendritic spine of analysis were traced back to their neuron of origin for the purposes of this study. Dendritic spines that could not be traced back to their neuron of origin were classified as non-traced and could belong to T5 cells or other cell types (Fig 1J). All synapses in this study correspond to manually proofread synapses.

## Allocation of polyadic synapse types

Apart from T5a, b, c and d subtypes, dendritic spines from Tm1, Tm9 and CT1 cells were found in Tm-, CT1-to-T5a connections. Such dendritic spines were not included in the allocation of polyadic T5 synapse types. Polyadic types were assigned qualitatively, regardless of the dendritic spine number (e.g., synapse type *abcd* can include 1a, 2b, 1c and 2d T5 spines).

## Spatial distribution of polyadic types

For the polyadic type distributions across the posterior-anterior T5a dendritic axis (Fig 2A), we measured the horizontal distance of every synapse from the root of the dendrite. The comparison was performed in the z-dimensional plane of the 3D Neuroglancer environment, as the most informative dimension for the posterior-anterior T5a dendritic extension (S1G Fig). The root was set manually as the first branching point of the dendrite.

## Lobula distribution of co-T5s

T5 subtypes residing at the same postsynaptic site with the T5a of analysis were named 'co-T5s'. For the spatial distribution of co-T5s, the horizontal distance of each co-T5 root from the T5a root of analysis was measured. For every T5a dendrite out of the five we analyzed, we evaluated the co-T5s per input type rather than per input cell (e.g., Tm1 connections in one T5a dendrite can consist of three Tm1 cells, as in S1F Fig). To exclude any bias from co-T5 overrepresentation due to connection strength, we counted each co-T5 once, regardless of the number of synapses it appeared into. We controlled for duplicates resulting from neuronal ID updates in FlyWire by examining the 3D reconstructions in Neuroglancer.

## Immunohistochemistry and confocal microscopy

Flies at early pupa stages were heat-shocked for 30 minutes at 37° to activate the Multi-color FlpOut cassette [34]. Fly brains (aged 2–5 days) were dissected in cold phosphate-buffered saline (PBS) and fixed in 4% paraformaldehyde (in PBS with 0.1% Triton X-100) for 24 minutes at room temperature, followed by three 10-minute washes in PBT (PBS and 0.3% Triton X-100). Brains were then incubated with primary antibodies in PBT for 48 hours at 4°C. After being incubated for 2 hours at room temperature, brains were PBT-washed four times for 15 minutes each and then incubated with secondary antibodies diluted in PBT for 48 hours at 4°C. After being incubated for 2 hours at room temperature and four 15 minutes PBT washes, brains were mounted with SlowFade Gold Antifade (Cat#S36936) for immediate sample viewing. Brains were imaged with a Leica Stellaris 5 laser scanning confocal microscope with a 63x glycerol immersive 1.3 NA HC Plan-apochromat objective at 2048 x 2048 x 0.4µm image resolution.

## Statistical analysis

Detailed analysis is reported in figure legends and was performed in GraphPad Prism v.9.3.0 and in Python v.3.9.18 with the use of seaborn 0.12.2, pandas 1.5.3, numpy 1.23.3 and scipy 1.9.3. Detailed information on the used statistical tests is reported in S2 Table.

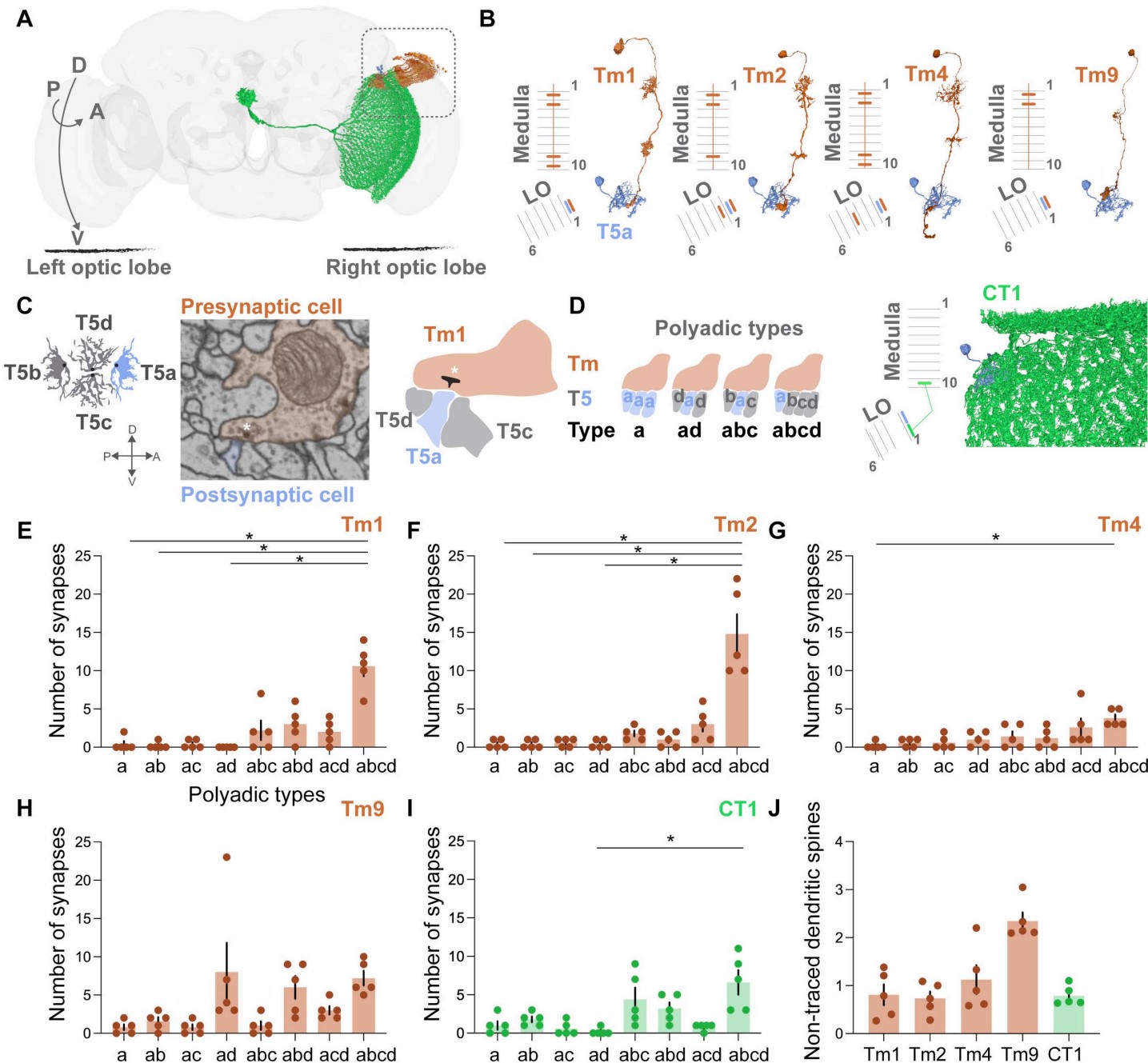

**Fig 1. Polyadic Tm1-, Tm2-, Tm4-, Tm9-, CT1-to-T5a synapses.** (A) Schematic representation of the circuit of analysis in the right optic lobe of the fruit fly brain (flywire.ai, S1 Table). (B) Dendritic ramifications in medulla layers and axonal terminals in lobula (LO) layers of Tm1, Tm2, Tm4 and Tm9 cells (orange). Dendritic ramifications in medulla layer 10 and lobula layer 1 (LO1) of CT1 cell (green). T5a dendrites in LO1 (blue). FlyWire IDs as in (A). (C) Schematic representation of T5 subtypes with the root denoted as a black circle (left). Electron microscopy snapshot of a polyadic Tm1-to-T5 synapse from flywire.ai (middle). Synapse coordinates: x = 181940, y = 46900, z = 4912. Asterisk indicates the presynaptic T-bar. Schematic representation of one presynaptic Tm1 bouton wiring to a T5d, T5a and T5c dendritic spine (right). (D) Schematic representation of polyadic Tm-to-T5 synapse types. (E-I) Number of *a*, *ab*, *ac*, *ad*, *abc*, *abd*, *acd* and *abcd* polyadic types in Tm1-, Tm2-, Tm4-, Tm9-, CT1-to-T5a synapses (nT5a=5). The normality of distribution was assessed with the use of Shapiro-Wilk test. Friedman test followed by Dunn's post hoc test where *p < 0.05. Data is mean±SEM. Detailed information is reported in S2 Table. (J) Average number of non-traced dendritic spines in Tm1-, Tm2-, Tm4-, Tm9- and CT1-to-T5a synapses of analysis (nT5a=5). Data is mean±SEM.

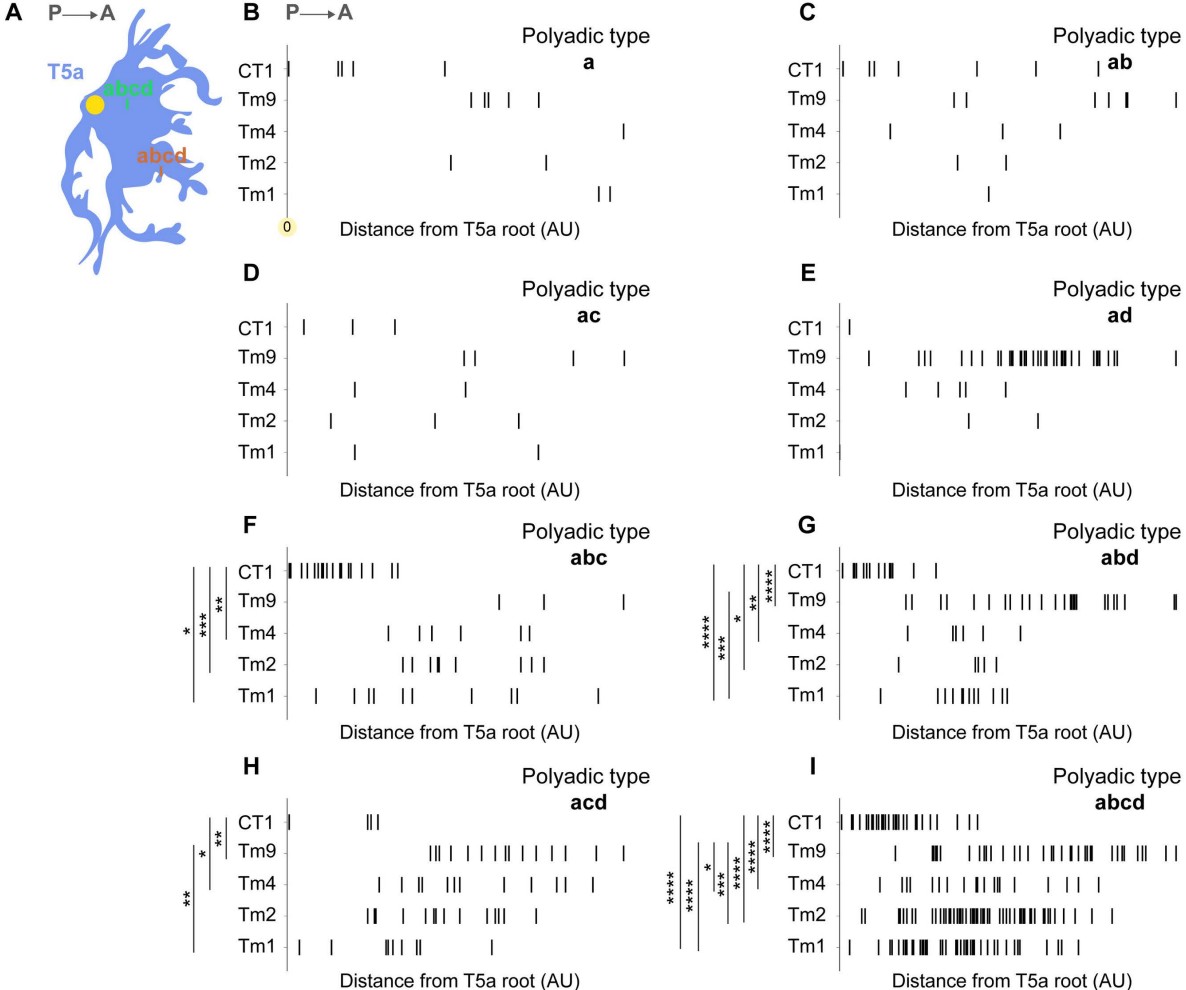

**Fig 2. Spatial organization of polyadic types on T5 dendrites.** (A) Schematic representation of *abcd* polyadic type between Tm9- (orange) and CT1-(green)-to-T5 connections on a T5a dendrite (blue). Horizontal distances of polyadic types were measured from the T5a dendritic root (yellow circle). (B) Horizontal distance of *a* polyadic type from the root of T5a dendrites (nT5a=5) across Tm1-, Tm2-, Tm4-, Tm9- and CT1-to-T5a synapses. (C-I) Same as in (B) for *ab*, *ac*, *ad*, *abc*, *abd*, *acd* and *abcd* polyadic types. The normality of distribution was assessed with the use of Shapiro-Wilk test. For normally distributed data a one-way ANOVA test followed by post-hoc pairwise t-tests with Bonferroni correction were used. For non-normally distributed data a Kruskal–Wallis test followed by Mann–Whitney U post hoc test with Bonferroni correction was used. *p < 0.05. **p < 0.01, ***p < 0.001, ****p < 0.0001. Detailed information is reported in S2 Table .

## Results

### Polyadic types in Tm1-, Tm2-, Tm4-, Tm9- and CT1-to-T5 connections

For our analysis, we used five, dorsally distributed, reconstructed T5a neurons as representatives of the horizontal motion detection system. Together with FlyWire annotations and previously attained morphological information [14–16,22], we identified the Tm1, Tm2, Tm4, Tm9 and CT1 cells connecting to T5a dendrites of analysis (Fig 1A and 1B). Polyadic synapses are characterized by a single presynaptic bouton with multiple postsynaptic compartments adjacent to it. Throughout our manual proofreading, we encountered Tm- and CT1-to-T5a synapses with multiple postsynaptic compartments, belonging to dendritic spines of different T5 subtypes (Fig 1C). Therefore, we allocated eight polyadic synapse types, namely *a*, *ab*, *ac*, *ad*, *abc*, *abd*, *acd* and *abcd* based on the T5 subtypes present in the

respective postsynaptic sites (Fig 1D). Importantly, all eight polyadic types were encountered across the five T5a neuronal inputs (Fig 1E–I). Additionally, we report certain complex synapse examples during the proofreading process. We treated two dendritic spines of the same T5a cell across one presynapse, spanning the same or sequential z-planes, as one polyadic synapse (S1A,B Fig). In two examples of multiple T-bars at the vicinity, we treated them as one and two synapses respectively based on their distinct synaptic clefts (S1 C,D Fig). Therefore, as in the larval *Drosophila* neuromuscular junction synapse (NMJ) [35,36], we too report multiple release sites (i.e., multiple T-bars) in Tm-to-T5a connections. Finally, the number of postsynaptic densities in Tm and CT1 boutons did not significantly vary across the five neuronal input types (S1E Fig). This excluded the regulation of synapse strength at the level of postsynaptic density abundancy.

Next, we asked if the Tm1-, Tm2-, Tm4-, Tm9- and CT1-to-T5 connections differ in terms of the polyadic types they use. The *abcd* type exhibited higher abundance compared to the other polyadic synapse types across the five neuronal inputs and was particularly enriched in Tm1 and Tm2 synapses (Fig 1E–I). However, polyadic type allocation is susceptible to dendritic spine reconstruction, hence the number of non-traced dendritic spines could affect our allocation. To tackle this, we calculated the average number of non-traced dendritic spines in all Tm and CT1 boutons across the T5a neurons of analysis (Fig 1J). Approximately one up to two dendritic spines were not traced in each postsynaptic density of analysis. Such error could result in quantitative differences, e.g., *abcd* type where a T5d spine is not traced, or qualitative differences, e.g., *abc* type where a T5d spine is not traced, both of which cannot be further resolved. In summary, visual information that is computed by a single Tm and CT1 neuronal bouton, could in principle be transmitted simultaneously to T5a, b, c, d neurons.

## Polyadic types introduce unique wiring patterns on T5 dendrites

The compartmentalized wiring of Tm1, Tm2, Tm4, Tm9 and CT1 cells on T5 dendrites has been extensively described at the morphological and functional level [15,21,22,25,37]. CT1 wires proximally to the dendrite root, Tm1, Tm2, Tm4 and Tm9 wire at the central dendritic compartment, while Tm9 extends to the most distal dendritic parts. This compartmentalized wiring will from now on be referred to as canonical wiring. Consequently, we wondered if the canonical wiring was recapitulated by the different polyadic synapse types (Fig 2A). The *a, ab, ac, ad* polyadic types did not mirror the canonical wiring, as the Tm9 extension to the distal dendritic site and the CT1 to the proximal were concealed, potentially due to the small sample size (Fig 2B–E, S2 Table). The *abc* and *acd* polyadic types preserved the CT1 extension to the proximal dendritic sites, while the *abd* and *abcd* types preserved both the Tm9-distal and CT1-proximal dendritic wiring (Fig 2F–I). Collectively, our results show that the canonical wiring of the various input neurons on T5a dendrites is not preserved across all polyadic types. It is thus possible that different polyadic types serve unique functional purposes in the computation of visual motion direction in T5a neurons.

## Polyadic types introduce unique wiring patterns in the T5 network

The polyadic types we previously described so far imply the simultaneous binding of neurotransmitters by transmitter receptors located on the dendrites of multiple T5 cells. We named the remaining T5 dendritic spines residing at the same postsynapse with the T5a spine of analysis as 'co-T5s' (Fig 3A, synaptic level). These co-T5 spines belong to T5 cells that are proximal to the T5a cell of analysis in the lobula and contribute to the T5 network (Fig 3A, columnar & neuropil level, Fig 3B). Therefore, we wanted to understand if there was an underlying wiring architecture imposed by the co-T5s. We focused on the posterior-anterior distribution of these co-T5s with respect to the T5a of analysis and calculated the horizontal distances between the T5a and co-T5 dendritic roots. In CT1-T5a polyadic connections, the co-T5s localized more on the anterior side of T5a dendrites in lobula layer one, while in Tm1-, Tm2-, Tm4-, Tm9-T5a polyads, the co-T5s localized more on the posterior side of T5a dendrites (Fig 3C–H). We conclude that CT1-derived co-T5 distributions significantly differ from the Tm1-, Tm2-, Tm4- and Tm9-derived ones.

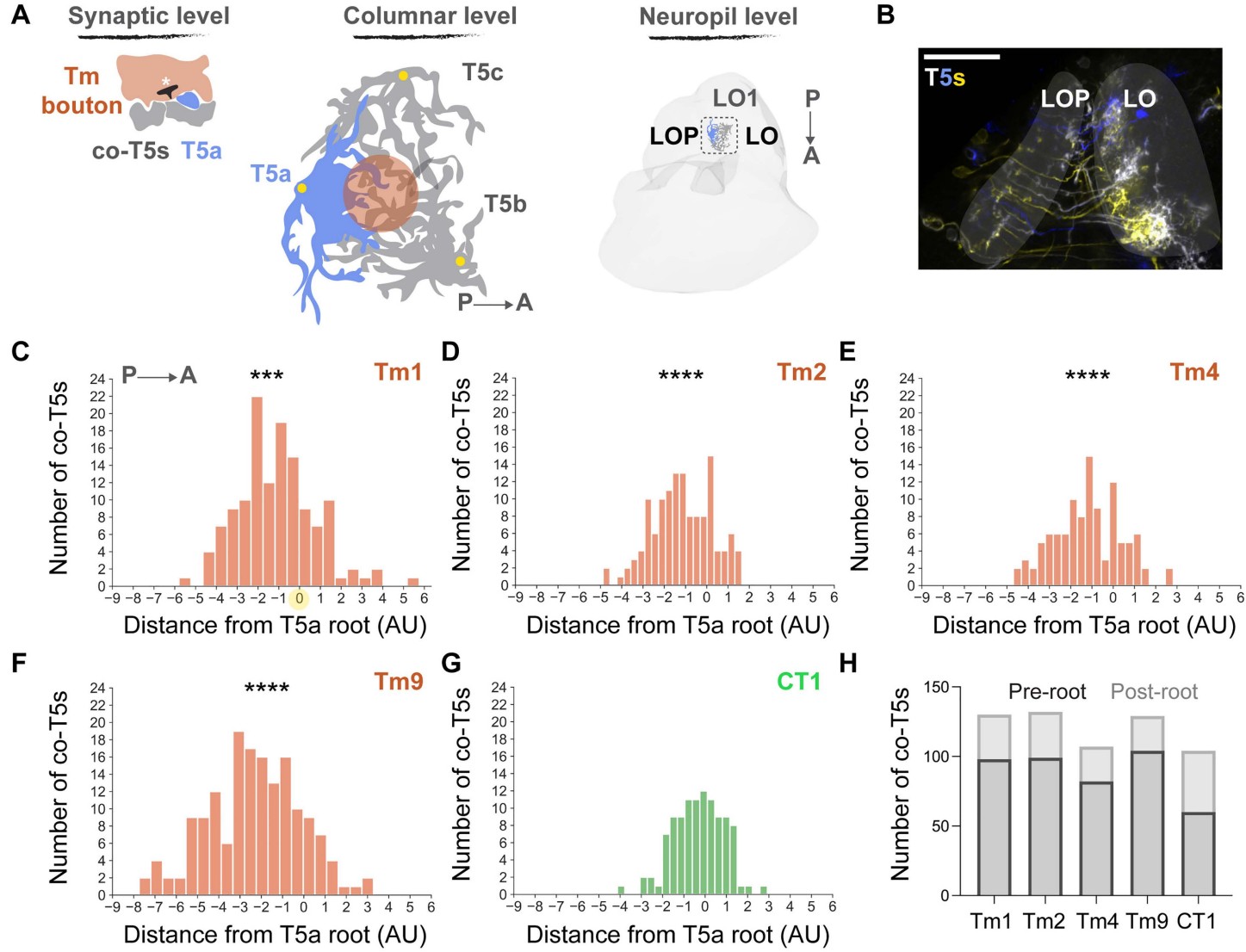

**Fig 3. Organization of co-T5s in lobula layer 1.** (A) Schematic representation of polyadic morphology from the synaptic to the neuropil level. Asterisk indicates the presynaptic T-bar. Dendritic roots in yellow circles. (B) Sparse T5 subtype labelling via the multi-color FlpOut approach. Scale bar 40μm. (C-G) co-T5 distribution from the root of T5a dendrites (nT5a=5) across Tm1- (nTm1 co-T5 = 132), Tm2- (nTm2 co-T5 = 132), Tm4-(nTm4 co-T5 = 107), Tm9-(nTm9 co-T5 = 161) and CT1-(nCT1 co-T5 = 105) to-T5a polyadic synapses. Dendritic root of T5a of analysis in yellow circle. The normality of distribution was assessed with the use of Shapiro-Wilk test. Two-tailed unpaired Student's t-test, where ***p < 0.001, ****p < 0.0001. Statistical test values correspond to Tm1-, Tm2-, Tm4-, Tm9-to CT1 comparisons. Detailed information is reported in S2 Table . (H) Number of co-T5s posterior (pre-root) and anterior (post-root) to the T5a root across Tm1-, Tm2-, Tm4-, Tm9- and CT1-to-T5a polyadic synapses.

To further understand the use of polyadic synapses in co-T5 distributions, we explored the co-T5 spatial organization deriving from the significantly abundant *acd* and *abcd* types in Tm1- and Tm2-to-T5 connections (Fig 1E–I). This analysis showed that *acd*-originating co-T5s followed a different spatial distribution compared to *abcd*-originating ones (S1H Fig). In conclusion, the co-T5 spatial embedding can be controlled at the input type and polyadic type level.

## Discussion

Even though the most abundant synapse morphology in *Drosophila melanogaster* is that of polyadic synapses, they remain understudied. Here, we report for the first time the wiring architectures that polyadic synapses introduce in the visual system of the fruit fly.

One of the prerequisites for direction-selective responses in T5 cells is the spatial offset between their neuronal inputs, accounting to the order of a lobula column. Tm1, Tm2, Tm4, Tm9 and CT1 used eight polyadic synapse types for their T5a connections, with *abc*, *abd*, *acd* and *abcd* types following the canonical wiring principles. Whether the remaining, diverging to the canonical wiring *a, ab, ac, ad* types introduce spatial differences equal to a full column length is not known. Therefore, advancements in defining columns and column size across the different neuropils in the optic lobe, as in the Visual Columns Map project (Codex), are needed.

One hypothesis is that apart from deviations in wiring patterns, each polyadic type could potentially introduce distinct functional profiles. Changes in neurotransmitter release probability depending on the postsynaptic partner have been previously observed in mammals [38,39], rendering the polyadic type allocation of our work a possible invertebrate example. Unique communication codes among polyadic types could be introduced by a suit of T5-specific or generalized mechanisms: differences in the type of acetylcholine receptors (AChRs) across the different T5 subtypes [22,40], the density and nanoarchitecture disparities observed across receptor types [41,42], the location of T5 dendritic spines compared to the presynaptic release site (transsynaptic nanocolumn, [43]), the neurotransmitter diffusion and recycling mechanisms [44,45] and the differences in neurotransmitter receptor numbers introduced by the total area differences of postsynaptic densities between polyadic types as observed in the rat hippocampus [46,47]. Further research with synaptic accuracy should be conducted to evaluate the different possibilities in T5 cells. Nevertheless, the inter-bouton AChR variety that was recently hypothesized in Tm9-to-T5 synapses [22], could derive from the different polyadic types we report in this work.

The polyadic morphology denotes the activation of postsynaptic T5 neurons at neighboring positions by the same presynaptic neuron, T5s that might not necessarily be visually activated. We found such 'co-T5s' from the GABAergic input neuron CT1 to be localized closer to the anterior side of the analyzed T5a cells in lobula, whereas those co-T5s postsynaptic to a single cholinergic Tm cell were found to localize closer to the posterior side of the analyzed T5a cells. This wiring architecture finding resembles an excitation-inhibition (E/I) polarization along the anterior-posterior axis. When a front-to-back motion stimulus activates the T5a neuron of interest, the majority of co-T5s posterior to that T5a would be excited and the majority of co-T5s anterior would be inhibited. If such a local feedforward adaptation mechanism were to be functionally verified, it could for example be used to compensate for contrast fluctuations [48,49]. What remains to be explored is whether the activation of co-T5s is synchronous, something that relies on the intricate diffusion mechanisms that govern each synapse.

Are polyadic types and their wiring motifs a unique trait of T5 cells and per extension of the OFF-motion pathway? A preliminary analysis in the horizontal system pointed toward their existence in chemical synapses between the GABAergic Mi4, C3 and the cholinergic Tm3 with T4b cells (S2A–E Fig). Hence, the ON-motion pathway shares the same polyadic type morphology. However, an excitation-inhibition (E/I) polarization along the anterior-posterior axis, similar to T5a cells, was absent in T4b neurons (S2F–S2I Figs). T4 cells have a greater variety of GABAergic inputs compared to T5 cells, therefore this discrepancy might be a distinct characteristic of CT1-to-T4 and CT1-to-T5 connections. Future studies should be performed in the totality of T4 inputs, so that novel comparative conclusions are drawn about T4 and T5 cell chemical connectivity.

In conclusion, polyadic synapses might play an important role in visual processing as well as in the development of the visual system. Our work represents a first step in that direction. With analyzing wiring diagrams of the brain and specifically of the optic lobe [1–4,50,51], we get insights on the fundamental principles that govern circuit architecture. Eventually, connectome analyses with single synapse connectivity resolution could lead to even more detailed single neuron

biophysical models [52]. By addressing connectivity architectures that emerge from the polyadic synapse morphology in the OFF-edge motion pathway, we propose a shift from the single neuron to the single synapse wiring diagram era.

## Supporting information

**S1 Fig. Synaptic proofreading in Tm1-, Tm2-, Tm4-, Tm9-, CT1-to-T5 synapses.** (A) Electron microscopy snapshot of one polyadic CT1-T5a synapse at a single z-plane (flywire.ai). Asterisks indicate the two T5a dendritic spines at the same CT1 presynaptic site. (B) Electron microscopy snapshot of one polyadic Tm2-T5a synapse at two sequential z-planes (flywire.ai). Asterisks indicate the two T5a dendritic spines at the same Tm2 presynaptic site. (C) Electron microscopy snapshot of one polyadic Tm1-T5a synapse at a single z-plane (flywire.ai). Asterisk indicates the three presynaptic T-bars. (D) Electron microscopy snapshot of two polyadic Tm1-T5a synapses at a single z-plane (flywire.ai). Asterisks indicate the two presynaptic T-bars. (E) Average number per neuronal input type of dendritic spines in Tm1-, Tm2-, Tm4-, Tm9- and CT1-to-T5a polyadic synapses of analysis. The normality of distribution was assessed with the use of Shapiro-Wilk test. Friedman test followed by Dunn's post hoc test. Data is mean±SEM. (F) Total number of input neurons per T5 dendrite of analysis. Data is mean±SEM. (G) T5a posterior-anterior dendritic distribution in the three-dimensional embedding in LO1 (flywire.ai). (H) abcd- (light orange) and acd- (orange) derived co-T5 distribution from the root of T5a dendrites (nT5a=5) across Tm1- and Tm2-to-T5a synapses. The normality of distribution was assessed with the use of Shapiro-Wilk test. Two-tailed unpaired Student's t-test, where ***$p < 0.001$. Detailed information is reported in S2 Table. (TIF)

**S2 Fig. Polyadic synapses in Mi4-, C3-, Tm3-to-T4 connections.** (A-C) Number of b, ab, bc, bd, abc, abd, bcd and abcd polyadic types in Mi4-, C3-, Tm3-to-T4b synapses (nT4b=3). Data is mean±SEM. (D) Total number of input neurons per T4 dendrite of analysis. Data is mean±SEM. (E) Average number per neuronal input type of dendritic spines in Mi4-, C3-, Tm3-to-T4b polyadic synapses of analysis. Data is mean±SEM. (F) T4b posterior-anterior dendritic distribution in the three-dimensional embedding in medulla (flywire.ai, T4b ID 720575940615711338). (G-I) co-T5 distribution from the root of T4b dendrites (nT4b=3) across Mi4-, C3-, Tm3-to-T4b polyadic synapses. Dendritic root of T4b of analysis in yellow circle. The normality of distribution was assessed with the use of Shapiro-Wilk test. Two-tailed unpaired Student's t-test, where *$p < 0.05$, **$p < 0.01$, ****$p < 0.0001$. Detailed information is reported in S2 Table. (TIF)

**S1 Table. Analyzed neuronal IDs from FlyWire (flywire.ai).** (DOCX)

**S2 Table. Detailed statistical analysis.** (DOCX)

## Author contributions

**Conceptualization:** Eleni Samara.

**Data curation:** Eleni Samara.

**Formal analysis:** Eleni Samara.

**Funding acquisition:** Alexander Borst.

**Investigation:** Eleni Samara.

**Methodology:** Eleni Samara, Alexander Borst.

**Project administration:** Alexander Borst.

**Resources:** Alexander Borst.

**Supervision:** Alexander Borst.

**Validation:** Eleni Samara.

**Visualization:** Eleni Samara.

**Writing – original draft:** Eleni Samara, Alexander Borst.

**Writing – review & editing:** Eleni Samara, Alexander Borst.

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
