## [Decision Letter · Decision Letter 0]

4 Sep 2025

We look forward to receiving your revised manuscript.

Kind regards,

Krishna Moorthi Bhat, M.D., Ph.D.

Academic Editor

PLOS ONE

Journal Requirements:

2. We noted in your submission details that a portion of your manuscript may have been presented or published elsewhere. As mentioned in our cover letter, this manuscript was part of my Doctoral Thesis and is available at the Ludwig Maximilian University of Munich (LMU) library and per extension the publication repository of the Max Planck Society (PuRe). Please clarify whether this [conference proceeding or publication] was peer-reviewed and formally published. If this work was previously peer-reviewed and published, in the cover letter please provide the reason that this work does not constitute dual publication and should be included in the current manuscript.

This work was funded by the Max Planck Society.

We are grateful to the Princeton FlyWire team and members of the Murthy and Seung labs for development and maintenance of FlyWire (supported by BRAIN Initiative grant MH117815 to Murthy and Seung). This work was funded by the Max Planck Society.

This work was funded by the Max Planck Society.

Additional Editor Comments :

Reviewer #1:

This manuscript addresses an important and understudied aspect of the Drosophila connectome - the role of polyadic synapses in shaping wiring architectures in the OFF-motion pathway, specifically in T5a neurons. The authors use FlyWire reconstructions to categorize polyadic synapse types and analyze their spatial distribution, introducing the concept of “co-T5s” as part of a directional wiring motif. This is potentially impactful, especially since polyadic synapses dominate the fly optic lobe but remain poorly understood. The manuscript is ambitious, bringing together connectomic analysis, structural categorization, and functional interpretation. However, while the findings are intriguing, the paper could be significantly strengthened by improvements in clarity, quantification, figure presentation, and the depth of discussion. Below, I outline major and minor comments.

Major Comments

1. The categorization of eight polyadic synapse types (a, ab, ac, etc.) is descriptive but risks appearing arbitrary unless functionally or structurally validated. The authors need to strengthen the justification for why this classification is meaningful beyond taxonomy. For example, do different types show systematic differences in abundance, PSD size, presynaptic bouton morphology, or co-T5 spatial distribution?

2. Figure 1 is strong overall, but the classification of polyadic types (a, ab, abc, abcd) needs clearer labeling of T5 subtypes and more consistent color coding to improve accessibility for readers outside the field.

3. Figure 2. While distributions and abundances are reported, the manuscript sometimes presents qualitative descriptions without statistical backing. Please provide statistical comparisons across polyadic types, beyond abundance counts. Are differences in spatial distribution significant across Tm vs. CT1 inputs? Include sample sizes, error bars, and test statistics. Also, the distance seems to be higher for a compared to abcd (Fig.B vs Fig.I), but according to Figure A, a is nearer than abcd.

4. The discussion presents intriguing hypotheses regarding E/I polarization, local feedforward adaptation, receptor heterogeneity, and potential coding strategies. However, these interpretations are currently speculative and may be overstated. Please explicitly label these as hypotheses, and connect them more directly to the structural data shown (rather than broader literature). Clarify what is supported by the data vs. what is extrapolation.

5. The study is limited to T5a in the OFF-pathway. Including at least a brief comparison with T4 neurons (ON-pathway) would significantly strengthen the paper. Even a preliminary check could reveal whether polyadic wiring architectures are a general feature or specific to OFF-motion circuits.

Minor Comments

1. The abstract is dense and highly technical. Consider simplifying the sentences and placing more emphasis on the main discovery—that polyadic synapses create directional wiring motifs in T5 neurons. Additionally, several terms are highly specialized and may only be familiar to readers within this specific field.

2. Introduction nicely contextualized with connectomics. However, the transition into the classification of polyadic types is abrupt. Adding a sentence on why categorizing synapse types matters for circuit function would help.

3. The naming of polyadic types (a, ab, abc, etc.) could be confusing to readers unfamiliar with T5 subtypes. A figure/table summarizing each type with schematic icons would help.

4. Phrases like “typical wiring” are somewhat vague. Consider defining “canonical/expected wiring” instead.

5. The claim that polyadic morphology “denotes synchronous activation” needs clarification. Could asynchronous release probabilities disrupt this? Please discuss alternative interpretations.

6. Ensure recent Drosophila visual connectome studies (2023–2025) are cited, especially those using FlyWire and Codex datasets.

7. Minor grammatical edits needed - “could in principle be simultaneously transmitted” to “could in principle be transmitted simultaneously”, “polyadic synapses… remain severely understudied” to “polyadic synapses… remain understudied”.

Reviewers' comments:

Reviewer's Responses to Questions

**Comments to the Author**

1. Is the manuscript technically sound, and do the data support the conclusions?

Reviewer #1: Partly

2. Has the statistical analysis been performed appropriately and rigorously?

Reviewer #1: No

3. Have the authors made all data underlying the findings in their manuscript fully available?

Reviewer #1: Yes

4. Is the manuscript presented in an intelligible fashion and written in standard English?

Reviewer #1: Yes

Reviewer #1: This manuscript addresses an important and understudied aspect of the Drosophila connectome - the role of polyadic synapses in shaping wiring architectures in the OFF-motion pathway, specifically in T5a neurons. The authors use FlyWire reconstructions to categorize polyadic synapse types and analyze their spatial distribution, introducing the concept of “co-T5s” as part of a directional wiring motif. This is potentially impactful, especially since polyadic synapses dominate the fly optic lobe but remain poorly understood. The manuscript is ambitious, bringing together connectomic analysis, structural categorization, and functional interpretation. However, while the findings are intriguing, the paper could be significantly strengthened by improvements in clarity, quantification, figure presentation, and the depth of discussion. Below, I outline major and minor comments.

Major Comments

1. The categorization of eight polyadic synapse types (a, ab, ac, etc.) is descriptive but risks appearing arbitrary unless functionally or structurally validated. The authors need to strengthen the justification for why this classification is meaningful beyond taxonomy. For example, do different types show systematic differences in abundance, PSD size, presynaptic bouton morphology, or co-T5 spatial distribution?

2. Figure 1 is strong overall, but the classification of polyadic types (a, ab, abc, abcd) needs clearer labeling of T5 subtypes and more consistent color coding to improve accessibility for readers outside the field.

3. Figure 2. While distributions and abundances are reported, the manuscript sometimes presents qualitative descriptions without statistical backing. Please provide statistical comparisons across polyadic types, beyond abundance counts. Are differences in spatial distribution significant across Tm vs. CT1 inputs? Include sample sizes, error bars, and test statistics. Also, the distance seems to be higher for a compared to abcd (Fig.B vs Fig.I), but according to Figure A, a is nearer than abcd.

4. The discussion presents intriguing hypotheses regarding E/I polarization, local feedforward adaptation, receptor heterogeneity, and potential coding strategies. However, these interpretations are currently speculative and may be overstated. Please explicitly label these as hypotheses, and connect them more directly to the structural data shown (rather than broader literature). Clarify what is supported by the data vs. what is extrapolation.

5. The study is limited to T5a in the OFF-pathway. Including at least a brief comparison with T4 neurons (ON-pathway) would significantly strengthen the paper. Even a preliminary check could reveal whether polyadic wiring architectures are a general feature or specific to OFF-motion circuits.

Minor Comments

1. The abstract is dense and highly technical. Consider simplifying the sentences and placing more emphasis on the main discovery—that polyadic synapses create directional wiring motifs in T5 neurons. Additionally, several terms are highly specialized and may only be familiar to readers within this specific field.

2. Introduction nicely contextualized with connectomics. However, the transition into the classification of polyadic types is abrupt. Adding a sentence on why categorizing synapse types matters for circuit function would help.

3. The naming of polyadic types (a, ab, abc, etc.) could be confusing to readers unfamiliar with T5 subtypes. A figure/table summarizing each type with schematic icons would help.

4. Phrases like “typical wiring” are somewhat vague. Consider defining “canonical/expected wiring” instead.

5. The claim that polyadic morphology “denotes synchronous activation” needs clarification. Could asynchronous release probabilities disrupt this? Please discuss alternative interpretations.

6. Ensure recent Drosophila visual connectome studies (2023–2025) are cited, especially those using FlyWire and Codex datasets.

7. Minor grammatical edits needed - “could in principle be simultaneously transmitted” to “could in principle be transmitted simultaneously”, “polyadic synapses… remain severely understudied” to “polyadic synapses… remain understudied”.

**Do you want your identity to be public for this peer review?** For information about this choice, including consent withdrawal, please see our Privacy Policy

Reviewer #1: No

---

## [Author Response · Author response to Decision Letter 1]

29 Sep 2025

Monday 29th of October 2025

Dear Dr. Krishna Moorthi Bhat and Dear Reviewer,

We would like to thank you for your feedback and the opportunity to revise and ameliorate our submitted work. We have now completed the journal and reviewer requirements, and hope that we have thoroughly answered all the points raised by the reviewer. We report all the changes below.

Journal Requirements

Our manuscript now meets the style requirements as per PLOS ONE instructions.

2. We noted in your submission details that a portion of your manuscript may have been presented or published elsewhere. As mentioned in our cover letter, this manuscript was part of my Doctoral Thesis and is available at the Ludwig Maximilian University of Munich (LMU) library and per extension the publication repository of the Max Planck Society (PuRe). Please clarify whether this [conference proceeding or publication] was peer-reviewed and formally published. If this work was previously peer-reviewed and published, in the cover letter please provide the reason that this work does not constitute dual publication and should be included in the current manuscript.

The first manuscript version that we submitted to PLOS ONE was embedded in my Doctoral thesis, which per university regulations became available at the Ludwig Maximilian University of Munich (LMU) library and the publication repository of the Max Planck Society (PuRe). Additionally, the manuscript was not formally peer reviewed or published in another scientific journal.

The dataset generated for this work is now publicly published in Zenodo in the following site: https://doi.org/10.5281/zenodo.17221018.

4. We note that the grant information you provided in the ‘Funding Information’ and ‘Financial Disclosure’ sections do not match. When you resubmit, please ensure that you provide the correct grant numbers for the awards you received for your study in the ‘Funding Information’ section.

The grant information provided in the ‘Funding Information’ and ‘Financial Disclosure’ now match.

5. Thank you for stating the following financial disclosure: This work was funded by the Max Planck Society. Please state what role the funders took in the study. If the funders had no role, please state: "The funders had no role in study design, data collection and analysis, decision to publish, or preparation of the manuscript." If this statement is not correct you must amend it as needed. Please include this amended Role of Funder statement in your cover letter; we will change the online submission form on your behalf.

For the mentioned financial disclosure, we now state the following: “This work was funded by the Max Planck Society. Support for ES and AB. The funders had no role in study design, data collection and analysis, decision to publish, or preparation of the manuscript.”

6. Thank you for stating the following in the Acknowledgments Section of your manuscript: We are grateful to the Princeton FlyWire team and members of the Murthy and Seung labs for development and maintenance of FlyWire (supported by BRAIN Initiative grant MH117815 to Murthy and Seung). This work was funded by the Max Planck Society. We note that you have provided funding information that is not currently declared in your Funding Statement. However, funding information should not appear in the Acknowledgments section or other areas of your manuscript. We will only publish funding information present in the Funding Statement section of the online submission form. Please remove any funding-related text from the manuscript and let us know how you would like to update your Funding Statement. Currently, your Funding Statement reads as follows: This work was funded by the Max Planck Society. Please include your amended statements within your cover letter; we will change the online submission form on your behalf.

We have now removed our acknowledgment section. Additionally, as per FlyWire recent guidelines, only a set of publications together with flywire.ai should be properly cited, hence the following acknowledgment was removed: “We are grateful to the Princeton FlyWire team and members of the Murthy and Seung labs for development and maintenance of FlyWire (supported by BRAIN Initiative grant MH117815 to Murthy and Seung)”.

Captions for our supporting information are now included at the end of our manuscript.

We proceeded with citing the suggested by the reviewer works.

Reviewer Requirements

Major Comments

1. The categorization of eight polyadic synapse types (a, ab, ac, etc.) is descriptive but risks appearing arbitrary unless functionally or structurally validated. The authors need to strengthen the justification for why this classification is meaningful beyond taxonomy. For example, do different types show systematic differences in abundance, PSD size, presynaptic bouton morphology, or co-T5 spatial distribution?

In Figure 1E-I, we show differences in the abundance of polyadic types which are input-type specific and at the same time we show that the qualitative attribution of polyadic types does not underline a quantitative difference in Figure S1E. Therefore, polyadic types introduce systematic differences in abundance between input types but not in the number of participating dendritic spines in each polyadic type. To assess the next two suggested morphological traits, PSD size and presynaptic bouton morphology, we would need to evolve new analysis tools and pipelines that would go beyond the purposes of our study. However, the co-T5 distribution in polyadic types can be currently assessed and for that we focused on the significantly abundant types, abd and abcd, in Tm1- and Tm2-to-T5 connections as per Figure 1E-I. We elaborate in lines 314-319 that the co-T5 distribution can be polyadic type-regulated and have included our new analysis as Figure S1H.

2. Figure 1 is strong overall, but the classification of polyadic types (a, ab, abc, abcd) needs clearer labeling of T5 subtypes and more consistent color coding to improve accessibility for readers outside the field.

We thank the reviewer for this suggestion. For the T5 subtype labelling, we incorporated a schematic in Figure 1C (left) where all four T5 subtypes and their respective dendritic directionality is visualized. Additionally, in Figure 1D we incorporated the schematic representation of the abc type, to have one representative per number of present subtypes, i.e. a-1 subtype, ad-2 subtypes, abc-3 subtypes, abcd-4 subtypes. Regarding the color-coding consistency, Tm cells are represented in orange, CT1 in green, T5as in blue and the remaining T5 subtypes in grey, a color code we follow throughout our work.

3. Figure 2. While distributions and abundances are reported, the manuscript sometimes presents qualitative descriptions without statistical backing. Please provide statistical comparisons across polyadic types, beyond abundance counts. Are differences in spatial distribution significant across Tm vs. CT1 inputs? Include sample sizes, error bars, and test statistics. Also, the distance seems to be higher for a compared to abcd (Fig.B vs Fig.I), but according to Figure A, a is nearer than abcd.

We thank the reviewer for this comment. We have now performed a detailed statistical analysis in Figure 2 which we extensively report (comparison, statistical test and significance, sample size) in the newly introduced Table S2 and in Figure 2 legend. As hypothesized by the reviewer, the statistical analysis revealed the Tm-to-CT1 differences in certain (abc, abd, acd, abcd) polyadic type distributions. We have adjusted our result section in lines 259-268 based on this new analysis. For the second part of this comment, we thank the reviewer for noticing this discrepancy, as the previous schematic in Figure 2A was abstract and not based on the results from Figure 2B-I. Now, we included a schematic representation of the abcd polyadic type between Tm9- (orange) and CT1-(green)-to-T5 connections on a T5a dendrite (blue) based on the Figure 2B-I results. Lastly, we incorporated the sample size information for Figure 3C-G as well.

4. The discussion presents intriguing hypotheses regarding E/I polarization, local feedforward adaptation, receptor heterogeneity, and potential coding strategies. However, these interpretations are currently speculative and may be overstated. Please explicitly label these as hypotheses, and connect them more directly to the structural data shown (rather than broader literature). Clarify what is supported by the data vs. what is extrapolation.

We thank the reviewer for this comment. We re-adjusted our discussion in lines 356-399 in the points below. We now clearly state in line 356 that the ‘distinct functional profiles’ of polyads is a hypothesis. In lines 357-359 we raise the possibility of polyadic types being analogous to the mammalian system in terms of changes in release probability introduced by postsynaptic partners, to more directly connect our structural data to the broader knowledge. However, we believe that it is essential to cite broader synaptic knowledge to show the different research possibilities that arise after allocating polyadic types. To align with the reviewer’s suggestion, we embedded in lines 368-369 the need to verify the speculations of polyadic function. The wording in lines 395-397 was also adjusted to comply with the reviewer’s suggestions. Lastly, in line 362 we cited another study -previously performed in our lab- where the different nicotinic AChR subunit expression across T5 subtypes is plotted (Fig.S1G), to directly connect our work with receptor heterogeneity, either type-based nicotinic vs muscarinic or nicotinic subunit heterogeneity.

5. The study is limited to T5a in the OFF-pathway. Including at least a brief comparison with T4 neurons (ON-pathway) would significantly strengthen the paper. Even a preliminary check could reveal whether polyadic wiring architectures are a general feature or specific to OFF-motion circuits.

We thank the reviewer for this suggestion. We have now proceeded with analyzing three T4b dendrites, to include both subtypes, i.e a and b, of the horizontal motion system across T4 and T5 cells. This addition is now included in supplementary Figure S2 and discussed in lines 401-423. In summary, we showed that polyadic types exist in T4 cells as well, but among the three analyzed dendrites no excitation/inhibition polarization was encountered as in T5 cells. Nevertheless, as mentioned in lines 407-421, the lack thereof can be the outcome of CT1-to-T4 connections, which due to the focus of our study in T5 cells did not proceed with their analysis. We do understand the need for comparison to assess the universality of the polyadic types in the two elementary motion detectors; however, we believe that our work is the essential first step towards that direction. The next steps would be to analyze all four subtypes across both T4 and T5 populations.

Minor Comments

1. The abstract is dense and highly technical. Consider simplifying the sentences and placing more emphasis on the main discovery—that polyadic synapses create directional wiring motifs in T5 neurons. Additionally, several terms are highly specialized and may only be familiar to readers within this specific field.

The abstract was adapted to the reviewer’s suggestions, excluding terms such as ‘preferred direction’ (line 17) which is highly technical. To point out the directionality that polyadic synapses introduce, we replaced the term complexity’ with directionality in line 23.

2. Introduction nicely contextualized with connectomics. However, the transition into the classification of polyadic types is abrupt. Adding a sentence on why categorizing synapse types matters for circuit function would help.

As suggested by the reviewer, we now added a justification on the importance of studying and categorizing morphological synaptic traits in lines 68-71.

3. The naming of polyadic types (a, ab, abc, etc.) could be confusing to readers unfamiliar with T5 subtypes. A figure/table summarizing each type with schematic icons would help.

We thank the reviewer for this suggestion. We added a schematic representation of T5 subtypes in Figure 1C (left) so that the reader better comprehends their morphological distinction. Moreover, in Figure 1D we introduced a schematic representation of the abc subtype, totaling to four polyadic type representations.

4. Phrases like “typical wiring” are somewhat vague. Consider defining “canonical/expected wiring” instead.

We have followed the reviewer’s suggestion and proceeded with replacing ‘typical’ with ‘canonical’ in the entirety of the result section ‘Polyadic types introduce unique wiring patterns on T5 dendrites’.

5. The claim that polyadic morphology “denotes synchronous activation” needs clarification. Could asynchronous release probabilities disrupt this? Please discuss alternative interpretations.

In Drosophila, the T-bar defines the presynaptic active zone and hence the presynaptic release sites. The majority of proofread presynapses comprises one T-bar: multiple T-bars are rarely found (as in Fig S1C). Therefore, the synchronicity would rely on diffusion mechanisms, depending on where each postsynaptic density resides, rather than on release probabilities. Following the reviewer’s suggestion, we removed the ‘synchronous’ in ‘synchronous activation’ in line 386 and introduced an explanation on the synchronicity in lines 397-399.

6. Ensure recent Drosophila visual connectome studies (2023–2025) are cited, especially those using FlyWire and Codex datasets.

In line 428 we cite the Matsliah et al., 2024 and Nern et al., 2025 studies.

7. Minor grammatical edits needed - “could in principle be simultaneously transmitted” to “could in principle be transmitted simultaneously”, “polyadic synapses… remain severely understudied” to “polyadic synapses… remain understudied”.

We incorporated these edits as suggested by the reviewer.

---

## [Decision Letter · Decision Letter 1]

5 Oct 2025

Polyadic synapses introduce unique wiring architectures in T5 cells of Drosophila

PONE-D-25-42365R1

Dear Dr. Samara,

We’re pleased to inform you that your manuscript has been judged scientifically suitable for publication and will be formally accepted for publication once it meets all outstanding technical requirements.

Kind regards,

Krishna Moorthi Bhat, M.D., Ph.D.

Academic Editor

PLOS ONE

Additional Editor Comments (optional):

Dear Dr. Samara,

I have had a competent reviewer look at your manuscript as well the revised version, and I myself have read the work, and am delighted to accept the work for publication in PLoS One. Congratulations and thank you for submitting your work to our journal.

Best regards,

Krishna M Bhat

Academic Editor

Reviewers' comments:

Reviewer's Responses to Questions

**Comments to the Author**

Reviewer #1: All comments have been addressed

2. Is the manuscript technically sound, and do the data support the conclusions?

Reviewer #1: Yes

3. Has the statistical analysis been performed appropriately and rigorously?

Reviewer #1: Yes

4. Have the authors made all data underlying the findings in their manuscript fully available?

Reviewer #1: Yes

5. Is the manuscript presented in an intelligible fashion and written in standard English?

Reviewer #1: Yes

Reviewer #1: The authors have made substantial revisions in response to my previous feedback. All of my comments have been satisfactorily addressed, and the manuscript has significantly improved in clarity, organization, and scientific rigor.

**Do you want your identity to be public for this peer review?** For information about this choice, including consent withdrawal, please see our Privacy Policy

Reviewer #1: No

---

## [Editor Report · Acceptance letter]

PONE-D-25-42365R1

PLOS ONE

Dear Dr. Samara,

I'm pleased to inform you that your manuscript has been deemed suitable for publication in PLOS ONE. Congratulations! Your manuscript is now being handed over to our production team.

Kind regards,

on behalf of

Professor Krishna Moorthi Bhat

Academic Editor

PLOS ONE